# Taurine Stimulates Thermoregulatory Genes in Brown Fat Tissue and Muscle without an Influence on Inguinal White Fat Tissue in a High-Fat Diet-Induced Obese Mouse Model

**DOI:** 10.3390/foods9060688

**Published:** 2020-05-26

**Authors:** Kyoung Soo Kim, Hari Madhuri Doss, Hee-Jin Kim, Hyung-In Yang

**Affiliations:** 1Department of Clinical Pharmacology and Therapeutics, Kyung Hee University School of Medicine, Seoul 02447, Korea; madhuridoss.h@gmail.com (H.M.D.); gmlwlskim2@khu.ac.kr (H.-J.K.); 2East-West Bone & Joint Disease Research Institute, Kyung Hee University Hospital at Gangdong, Gandong-gu, Seoul 05278, Korea; yhira@khu.ac.kr

**Keywords:** taurine, brown fat tissue (BAT), inguinal white fat tissue (iWAT), high-fat diet (HFD), thermogenesis, anti-obesity

## Abstract

This study was conducted to investigate if taurine supplementation stimulates the induction of thermogenic genes in fat tissues and muscles and decipher the mechanism by which taurine exerts its anti-obesity effect in a mildly obese ICR (CD-1^®^) mouse model. Three groups of ICR mice were fed a normal chow diet, a high-fat diet (HFD), or HFD supplemented with 2% taurine in drinking water for 28 weeks. The expression profiles of various genes were analyzed by real time PCR in interscapular brown adipose tissue (BAT), inguinal white adipose tissue (iWAT), and the quadriceps muscles of the experimental groups. Genes that are known to regulate thermogenesis like PGC-1α, UCP-1, Cox7a1, Cox8b, CIDE-A, and β_1_-, β_2_-, and β_3_-adrenergic receptors (β-ARs) were found to be differentially expressed in the three tissues. These genes were expressed at a very low level in iWAT as compared to BAT and muscle. Whereas, HFD increased the expression of these genes. Taurine supplementation stimulated the expression of UCP-1, Cox7a1, and Cox8b in BAT and only Cox7a1 in muscle, while there was a decrease in iWAT. In contrast, fat deposition-related genes, monoamine oxidases (MAO)-A, and -B, and lipin-1, were decreased by taurine supplementation only in iWAT and not in BAT or muscle. In conclusion, the potential anti-obesity effects of taurine may be partly due to upregulated thermogenesis in BAT, energy metabolism of muscle, and downregulated fat deposition in iWAT.

## 1. Introduction

Taurine (2-aminoethane-sulfonic acid) is a sulfur-containing amino acid and is regarded as the most abundant free amino acid in a number of mammalian tissues [1]. The intracellular concentration of taurine varies at 2–20 μmole/g wet weight in many tissues such as the brain, heart, and skeletal muscles [1,2]. Taurine is endogenously synthesized from cysteine and methionine. Primarily synthesized in the liver and kidneys, taurine is used for bile acid conjugation and to counter a significant change in osmotic pressure, respectively [1,3,4]. Other tissues are known to partly contribute to its synthesis depending on the presence of key enzymes such as cysteine dioxygenase (CDO) and cysteinesulfinic acid decarboxylase (CSDA) even if tissue synthesis is lower compared to the amount in the liver and kidneys. Unexpectedly, the mRNA expression of the CDO enzyme, taurine synthesis in the epididymal, and perirenal white adipose tissues of rats appears to be similar to that of the liver and kidneys [5]. Therefore, taurine is actively synthesized in white adipose tissue, but adipose taurine synthesis is reduced in obese mice [6,7]. These results suggest an important physiological role of taurine in adipose tissue as well as in the liver and kidneys [7,8]. 

Among the various beneficial effects of taurine [8,9], the anti-obesity effect has been analyzed in clinical studies [10,11,12] as well as in animal models [13,14,15,16]. Many studies have indicated that taurine deficiency is associated with metabolic dysfunction, such as obesity or diabetes [17,18], and its supplementation may help slow down the progression of obesity. Recently, it was reported that taurine supplementation improved oxidative stress indices and inflammatory biomarkers in patients with type 2 diabetes mellitus [19]. These results show the potential taurine possesses to be developed as a safe and beneficial agent to reduce obesity-associated metabolic dysfunctions such as dyslipidemia, insulin resistance, and hyperglycemia. The anti-obesity effect of taurine has been well demonstrated in animal studies. However, the dose of taurine used to treat obese animals has not been physiologically practical to be included as a dietary regimen for humans. This may be one of the reasons why the anti-obesity effects of taurine have not been clearly shown in clinical human studies [20]. Also, the C57BL/6 mice used as an obesity model is known to develop severe obesity when fed a high fat diet (HFD) due to their genetic susceptibility to diet-induced obesity (DIO) [21]. Even though C57BL/6 and ICR mice overall have similar physiological and metabolic phenotypes [22], C57BL/6 HFD fed mice also develop dyslipidemia and changes in glucose homeostasis resembling a pre-diabetic condition, whereas ICR mice show mild obesity without any pre-diabetic conditions. Therefore, ICR HFD fed mice may be a more suitable model system for mimicking mild obesity similar to humans than C57BL/6 mice. In our previous study, we developed a mild obese model using HFD-fed ICR mice and demonstrated the anti-obesity effects of 2% taurine supplemented drinking water [23]. In this study, we showed one of the molecular mechanisms by which taurine ameliorates mild obesity. Taurine supplementation downregulated the expression of adipogenesis-related genes such as PPAR-α, PPAR-γ, C/EBP-α, C/EBP-β, and AP2 in white adipose tissue (WAT) but not in brown adipose tissue (BAT) [23]. BAT is responsible for heat generation (thermogenesis) through a process called uncoupled respiration mediated by uncoupling protein-1 (UCP1), while WAT stores energy [24,25]. Taurine supplementation increased body temperature in a monosodium glutamate (MSG)-induced obesity rat model as compared to obese control rats [15]. Taurine treatment decreased the weight of WAT and increased the weight of BAT by inducing the expression of peroxisome proliferator-activated receptor gamma co-activator 1α (PGC-1α), which is involved in stimulation of energy expenditure. All these reports suggest that taurine may enhance the “browning” process in adipocytes and induce thermogenesis from BAT by increasing the expression of genes related to energy expenditure. Browning is the process that increases the number of beige or brite (brown in white) adipocytes, which help to increase energy expenditure and reduce obesity through thermogenic heat generation [26]. Therefore, augmentation of brown fat mass and/or its activity has become a promising strategy to reduce obesity without side effects [27]. As a result, many researchers have aimed to discover dietary compounds that increase the browning process or energy expenditure [28,29,30] and also tried to evaluate which genes could be markers of beige adipocytes in WAT [26]. Other thermoregulatory genes such as Cytochrome C Oxidase Subunit VIIIb (Cox8b), Cox7a, and Cell Death-Inducing DFFA-Like Effector A (CIDEA) were also specific markers of beige adipocytes. Also, BAT is densely innervated by the sympathetic nervous system (SNS), which governs its thermogenesis. Catecholamines released from sympathetic neurons stimulate β-adrenergic receptors present on the cell surface, ultimately activating UCP-1-dependent thermogenesis [31]. In addition, catecholamine such as dopamine, epinephrine (adrenaline), and norepinephrine are known to activate lipolysis, mainly through β-adrenergic receptor activation [32]. The level of catecholamine is regulated by the monoamine oxidase (MAO) enzyme and inhibition of MAO results in loss of body weight [33]. In addition, adipogenesis-related increase of monoamine oxidase (MAO) is known to occur in adipocytes. Therefore, MAO may be a target for therapeutic intervention in obesity [34]. Furthermore, lipin-1 of the lipin family plays a key role in lipid synthesis due to its phosphatidate phosphatase activity and also because it acts as a transcriptional co-activator to regulate the expression of genes involved in lipid metabolism [35]. In this study, we investigated how taurine affects the expression of thermogenesis- and browning-related genes in muscles, WAT, and BAT in a mild obese mouse model.

## 2. Materials and Methods

### 2.1. Animals and Diets

Thirty male, four-week old ICR (CD-1^®^) mice were randomly subdivided into three groups, housed in a specific pathogen-free (SPF) facility with a 12 h light/dark cycle, and given *ad libitum* access to food and water. As described previously [23], for 28 weeks, the first group was fed a normal chow diet and it was named as the Normal group (*n* = 10), the second group was fed a high-fat diet (HFD) and named as the HFD group (*n* = 10), and the third group was fed HFD supplemented with 2% taurine in the drinking water and was named the HFD + TAU group (*n* = 10). All animal protocols were approved by the Committee on Animals of Kyung Hee University Hospital at GANGDONG (KHNMC AP 2016-009). Nara Biotech (Seoul, Korea) provided the HFD, Research Diets D12451 diet (45 kcal% fat). Dong-A Pharmaceuticals (YongIn, Korea) supplied the taurine. The HFD + TAU group received taurine in purified drinking water containing 2% taurine, while unadjusted purified water was provided to the Normal and HFD groups.

### 2.2. Food Uptake, Activity, and Metabolic Parameters

As described previously [23], mouse body weight was monitored weekly for 28 weeks. Metabolic monitoring was assessed in a resting state using the PhenoMaster System (TSE systems GmbH, Bad Homburg, Germany). Energy expenditures including CO_2_ production (VCO_2_) and O_2_ consumption (VO_2_) were monitored for 48 h. The mice were free to consume food and water. The respiratory exchange ratio (RER) was defined as the ratio of carbon dioxide volume versus oxygen volume (VCO_2_/VO_2_). Food uptake and locomotor activity were also measured. An LF50 body composition analyzer (Bruker, Germany) was used to determine body composition (lean body mass, total body fat, and fluid) in mice. Animals were given 4–6 h to acclimate to the metabolic caging prior to beginning data collection, which took place over a 24 h period. Data collected (respiratory exchange ratio (RER), VO_2_, VCO_2_, energy expenditure (EE), food uptake, drinking, and activity) were separately averaged over the light and dark periods. Animals were maintained on a 12 h light–dark cycle, continued to consume a standard rodent chow diet, and were provided with water *ad libitum*. All procedures were approved and ethical consent was provided by the Animal Care Committee at Seoul University College of Veterinary Medicine, Korea Mouse Phenotyping Center (KMPC).

### 2.3. Body Weight and Composition

Mouse body weight was monitored every week for 28 weeks. An LF50 body composition analyzer (Bruker Co., Billerica, MA, USA) was used to determine body composition (lean body mass, total body fat, and fluid) in mice. This analyzer is based on Time Domain Nuclear Magnetic Resonance (TD-NMR) technology. The animal was loaded into the sample holder (animal restrainer not to limit movement of the animal). The animal holder was inserted into the instrument LF50 for analysis. Results are displayed and stored on a PC.

### 2.4. Harvest Tissues from Mice

As described previously [23], inguinal white adipose tissue, interscapular brown adipose tissue, and quadriceps muscles were harvested from euthanized mice by cervical dislocation, instantly frozen in liquid nitrogen, and kept at −80 °C until analysis. Total RNA was extracted from the harvested tissues using Trizol (Thermo Fisher Scientific Korea, Seoul, Korea).

### 2.5. Quantitative Real-Time RT-PCR

cDNA was synthesized from RNA using a commercial cDNA synthesis kit (Thermo Fisher Scientific Korea, Seoul, Korea) according to the manufacturer’s instructions. Quantitative real-time RT-PCR was performed using an Applied Biosystem™ Real-Time PCR system (Applied Biosystems, Carlsbad, CA, USA) with the primer sequences as shown in Table 1. The relative mRNA expression of the target gene was calculated using the ΔΔCt method and was normalized to 18S rRNA as an internal control.

### 2.6. Statistical Analysis

Experimental data are expressed as mean ± standard error of the mean (SEM). Differences between the three groups were analyzed using the nonparametric Kruskal-Wallis test. If a statistical difference was detected (*p* < 0.05), post-hoc pairwise group comparisons were performed using Dunn’s test with Bonferroni multiple-testing correction [36]. Prism software v.5 (Graphpad Software, San Diego, CA, USA) was used for statistical analysis and generating graphs. Differences were considered statistically significant at *p* < 0.05.

## 3. Results

### 3.1. Effect of Taurine on Anti-Obesity in HFD-Induced Mildly Obese ICR Mice

To prepare the mild obesity mouse model, ICR mice were fed an HFD for 28 weeks to mimic mild human obesity. As shown previously [23], HFD significantly increased body weight over 28 weeks of feeding compared to mice receiving a normal diet. Taurine supplementation (2% in drinking water) significantly inhibited body weight gain over 28 weeks (Figure 1A). The mean body weights (mean ± SEM) of HFD-fed mice and normal mice were 55.90 ± 2.71 g and 45 ± 1.21 g, respectively. Taurine supplementation (2% in drinking water) in HFD-fed mice induced weight loss in HFD-induced mildly obese ICR mice compared to HFD-fed mice (55.9 ± 2.71 g vs. 49.33 ± 1.13 g). In accordance with the body weight change, body composition analyzer showed that the fat mass of HFD-fed mice significantly increased to 25.28 ± 1.22% of body weight from 9.74 ± 1.22% of the normal diet-fed group (Figure 1B). Taurine supplementation reversed the fat mass build up to 11.69 ± 5.25%. Taken together, these findings suggest that long-term taurine supplementation (2% in drinking water) may result in the loss of fat mass in HFD-fed mice.

### 3.2. Effect of Taurine on the Transcriptional Expression of Thermogenesis-Related Genes in Fat Tissue and Muscles of HFD-Induced Mildly Obese ICR Mice

First, the transcriptional expression levels of several thermoregulatory and beige cell marker genes such as PGC-1α, UCP-1, Cox7a1, Cox8b, and CIDE-A were investigated and compared in three tissues: inguinal white adipose tissue (iWAT), interscapular brown adipose tissue (BAT), and quadricep muscles (Muscle). As shown in Figure 2A, all the genes were highly expressed in BAT, but not in iWAT at the mRNA level. The lower level expression of these genes in WAT suggested that they were associated with energy expenditure and were not required for WAT functioning of energy storage. In contrast, the expression levels of PGC-1α, Cox7a1, and Cox8b in muscles were as high as those seen in BAT, suggesting that energy expenditure is required for muscle movement. Interestingly, neither UCP-1 nor CIDE-A were highly expressed in muscle and were as low as expressed in iWAT. Next, we investigated how HFD or taurine supplementation could change the gene expression level. As shown in Figure 2B, HFD significantly stimulated the expression of genes such as PGC-1α and UCP-1 in BAT and slightly increased in the three tissues. In addition, taurine supplementation further stimulated gene expression in BAT and muscle.

### 3.3. Effect of Taurine on the Transcriptional Expression of β_1, 2, 3_-Adrenergic Receptors in Fat Tissue and Muscles of HFD-Induced Mildly Obese ICR Mice

To investigate if taurine affects the expression of β-adrenergic receptors involved in browning and thermogenesis [37,38], the basal mRNA expression levels of three subtypes of β_1, 2, 3_-adrenergic receptors (β_1_; *ADRB1*, β_2_; *ADRB2*, and β_3_; *ADRB3*) were compared in the three tissues. As shown in Figure 3A, all the receptors were highly expressed in BAT, while the expression of the β_2_-adrenergic receptor (*ADRB2*) at high basal level was only found in muscle. In contrast, iWAT expresses the three types of receptors at very low levels compared to those of BAT and muscle. *ADRB3* expression in iWAT was higher than the other two subtypes. As shown in Figure 3B, HFD slightly increased the expression of *ADRB1* and *ADRB3* in BAT, but only affected *ADRB2* expression in iWAT. In contrast, only *ADRB3* was significantly increased in muscle by HFD and taurine supplementation. All these results suggest that HFD or taurine-mediated β-adrenergic receptors are not significantly involved in thermogenesis in BAT and iWAT, but that taurine could have the potential to stimulate thermogenesis in muscle through stimulation of ADRB3 expression.

### 3.4. Effect of Taurine on Transcriptional Expression of Monoamine Oxidases (MAOs) and Lipin-1 in Fat Tissue and Muscles of HFD-Induced Mildly Obese ICR Mice

To investigate if taurine affects the expression of MAOs and lipin-1 involved in thermogenesis or lipid content, the basal expression level of three genes is checked (Figure 4). The expression of MAO-A is expressed in three tissues at similar levels, but the expression of MAO-B and lipin-1 in iWAT is much lower than that in BAT and muscle (Figure 4A). Also, taurine and HFD did not greatly affect the expression of MAO-A, MAO-B, and the lipin-1 gene in BAT and muscle (Figure 4B). In contrast, HFD increased the expression of the three genes in iWAT, but taurine supplementation significantly decreased the upregulated expression of these genes. These results indirectly suggest that taurine supplementation partly contributes to the anti-obesity effect via the inhibition of lipid synthesis in iWAT but not in BAT or muscle.

### 3.5. Effect of Taurine on Infiltration of Macrophages into Fat Tissues

Next, we checked if taurine supplementation inhibits the infiltration of M1 and M2 macrophages because alternatively activated M2 macrophages can control BAT thermogenesis through the local release of catecholamine [39]. As shown in Figure 5A, as expected, the basal transcriptional levels of both macrophage marker genes (F4/80) and M2 macrophage-specific marker genes (CD206 and CD163) were higher in iWAT than that of BAT and muscle. In addition, the expression of M1 macrophage marker genes (CD11c and CD68) in iWAT was significantly higher than in BAT and muscle. This means that macrophages remain more in iWAT than in BAT and muscle. However, HFD increased the level of F4/80 in BAT as well as iWAT, suggesting that more macrophages infiltrated into increased adipose tissues, even though all the marker genes such as M2 macrophage marker genes (CD206 and CD163) as well as M1 macrophage marker genes (CD11c and CD68) were not significantly increased in iWAT and BAT (Figure 5B). This suggests that HFD induced the infiltration of more macrophages into increased fat tissues. In contrast, taurine supplementation significantly decreased the macrophage marker gene expression (F4/80) in iWAT but had no effect in BAT. This indirectly suggests that taurine supplementation inhibits the increase of iWAT more than BAT by HFD feeding.

## 4. Discussion

The functional role of dietary taurine in reducing obesity has been reported earlier. However, there are not many studies that document the exact underlying molecular mechanism of its anti-obesity effects. In our previous study, it was reported that taurine mediated the inhibition of HFD-induced adipogenesis in iWAT, but not in BAT, while taurine did not affect the HFD uptake for 2 days in a metabolic cage. This prompted us to investigate if taurine supplementation induced weight loss was a result of up-regulated thermogenic mechanisms including browning of iWAT and via activation of BAT. It was our intention to check the expression of thermoregulatory genes such as PGC-1α, UCP-1, Cox7a1, Cox8b, and CIDE-A in BAT and iWAT involved in this process [40,41].

As shown in Figure 2, only HFD intake stimulated the expression of these genes in BAT and the iWAT of this study. This can be explained by the fact that energy expenditure is triggered in response to caloric excess. HFD induces acquisition of brown adipocyte-associated gene expression features in white adipose tissue [42]. Also, diet-induced thermogenesis occurs in brown adipose tissue (BAT) through the increase of UCP1 expression [43]. However, taurine treatment in our study decreased the expression of PGC-1α, Cox7a1, Cox8b, and UCP-1, which was increased by HFD in iWAT, but not in BAT. Meanwhile, another recent study showed that taurine exerts anti-obesity effects. The molecular mechanism was explained by browning WAT through the significantly elevated expression of PGC-1α and UCP-1 in iWAT [44]. This result was opposite to our result. This discrepancy might be partly due to the differences in the genetic background based on diet-induced obesity (DIO) between C57BL/6 and ICR mice. Extending support to this thought, obesity-prone C57/BL6 and obesity-resistant SV129 mice have differential expression of UCP-1 level in subcutaneous fat, which could mediate obesity resistance [45]. Thus, this indirectly suggests that in a mild obese model of ICR mice, taurine-mediated thermogenesis in BAT could contribute more to weight loss than taurine-mediated browning in iWAT. Meanwhile, UCP-1 is mainly expressed in BAT, while UCP-3 expression is largely expressed in skeletal muscle mitochondria [46]. Thus, we wondered how UCP-3 expression is changed in response to HFD or taurine supplementation. UCP-3 expression in muscle, BAT, and iWAT is changed similarly to the UCP-1 expression pattern in response to taurine (data not shown). Thus, it was thought that the taurine-mediated decreased expression of PGC-1α and UCP-1 in iWAT is not an artefact but could be induced in obesity-resistant ICR mice. Furthermore, the cytochrome c oxidase subunit isoform Cox7a1 and Cox8b is highly abundant in skeletal muscle and is well-known as brown adipocyte gene marker, as a cold-responsive protein of brown adipose tissue [47]. The expression of two genes was also changed with the same pattern in response to taurine. Also, CIDE-A (cell death-inducing DNA fragmentation factor α-like effector A) is highly expressed in thermogenesis-competent adipose cells such as brown and brite. As shown in Figure 2, the gene was highly expressed in BAT compared to that of iWAT and muscle. The CIDE-A expression was also reduced in iWAT by taurine supplementation.

The thermogenic regulation in BAT has been strongly associated with activation of α and β adrenergic receptors (AR) in the SNS by norepinephrine (NE) [38]. HFD-mediated AR activation stimulates lipolysis and energy expenditure [48,49]. HFD also activates β-AR [50]. The three subtypes of β-AR, β1, β2, and β3 (*ADRB1*, *ADRB2*, and *ADRB3*) in adipose tissue are differentially expressed at a ratio of 3:1:150 in mice at normal temperatures. Higher expression of *ADRB3* results in a much lower affinity for catecholamines than with *ADRB1* and *ADRB2* [51]. These subtypes are also able to regulate the expression of the UCP gene family [52]. Disruption of *ARDB3* increases susceptibility to diet-induced obesity in mice [53]. Therefore, *ADRB3* seems to be more important for BAT activation, energy expenditure, lipolysis, and reduced fat mass. Therefore, in this study, HFD induced higher expression of *ADRB3* than *ARDB1* and *ADRB2* in BAT, but not in iWAT. Taurine also significantly stimulated the expression of ADRB3 in muscle, giving insight into developing novel targets to pharmacologically activate energy expenditure in muscle. Meanwhile, ARDB2, which is highly expressed in skeletal muscle, plays a critical role in the maintenance of muscle mass by enhancing cAMP signaling through the activation of ADRB2. Several food factors are known to show agonistic activity at mouse or human ADRB2 [54]. Taurine also has potential to modulate the expression of ADRBs in tissues. On the contrary, with the activation of ADRBs by taurine, this could inhibit the release of acetylcholine and norepinephrine at synapses as a modulator of neuronal activity [55]. Also, it is beneficial in chronic heart failure (CHF) as it deactivates the sympathetic nervous system through inhibition of catecholamine [18]. Thus, taurine may have no significant effect on ADRB-mediated thermogenesis because it downregulates the release of catecholamine, even though it greatly stimulated the expression of ADRB3 in muscle in this study.

Also, catecholamines such as dopamine, norepinephrine, and serotonin are well known to be associated with emotion. Thus, decreased levels of these three neurotransmitters have been linked with depression and anxiety [56,57]. Monoamine oxidase inhibitors (MAOIs) are useful at relieving symptoms associated with depression, such as sadness or anxiety, because they block the actions of MAO in the brain [58]. Taurine also showed potential antidepressant activity in chronic unpredictable mild stress-induced depressive rats. It was explained that taurine may be involved in the regulation of the hypothalamic-pituitary-adrenal (HPA) axis and the promotion of neurogenesis, neuronal survival, and growth in the hippocampus [59]. However, in our study, taurine supplementation inhibited the expression of MAO in iWAT and thus, taurine-mediated MAO inhibition in the brain partly acted as an anti-depressant in the rat model. This suggests that taurine indirectly induces the change of catecholamine, but that taurine-mediated MAO expression does not significantly contribute to weight loss.

Resident adipose tissue macrophages (ATMs) respond to the drastic changes in adipose tissues under obesity conditions. Obesity induced low-grade inflammation causes a phenotypic shift in ATMs from so-called alternatively activated “M2” macrophages to classically activated “M1” macrophages in adipocyte niche through interplay with other immune cells [60]. In this study, taurine inhibited HFD-mediated infiltration of M1 macrophages but could have the potential to modestly affect macrophage polarization from M1 to M2, which may be due to weak inflammatory conditions in our mild obese model. Also, M1 macrophage induces insulin resistance through the production of pro-inflammatory cytokines and suppresses the induction of thermogenic adipocytes through inhibition of UCP-1 expression in obese adipose tissues [61]. M2 macrophages contribute to thermogenesis through activation of *ADRB3* by release of catecholamine [39]. Thus, it should be considered that taurine-mediated macrophage activation contributes to body weight loss.

Recent investigations have demonstrated that dietary supplements containing thermogenic constituents can increase resting metabolic rate (RMR) and thereby also promote thermogenesis through BAT activation and beige fat development [62,63]. Dietary constituents contain many phytochemicals (e.g., capsaicin, resveratrol, curcumin, green tea, and berberine), dietary fatty acids, and trans retinoic acid, a vitamin A metabolite. Flavonoids, potential bioactive compounds, are also suggested to activate non-shivering thermogenesis [29]. Furthermore, “non-energetic food constituents” such as smell and taste through sensory nerve stimulation have been found to be intrinsically linked with the accelerated expression of diet-induced thermogenesis, which accompanies the burning of fat within brown adipose tissues (BAT) [28]. These dietary factors have received more attention as a promising solution to increase energy expenditure. They have potential as therapeutic agents that ameliorate obesity by activating or inducing BAT and UCP1.

## 5. Conclusions

In conclusion, the data indirectly provide insights to explore the molecular mechanisms by which taurine exerts its anti-obesity effect. Our study suggests that these effects are a result of taurine induced activation of thermogenesis in BAT and inhibition of fat deposition in WAT. Long-term taurine supplementation in a mildly obese ICR mouse model induced weight loss. However, regulated clinical trials are necessary to determine the anti-obesity effects of taurine in humans.

## Figures and Tables

**Figure 1 foods-09-00688-f001:**
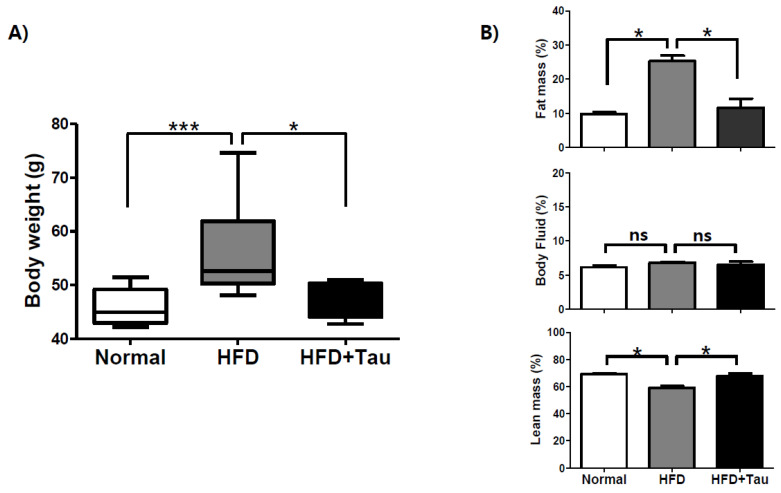
Effect of taurine on body weight loss in high-fat diet (HFD)-fed ICR mice. (**A**) ICR mice were fed a normal chew diet, HFD, or HFD + taurine (2% in drinking water) for 28 weeks (*n* = 10/group). They were named as Normal group, HFD group, and HFD + Tau group, respectively. (**B**) Body composition of the three groups (*n* = 4) was analyzed with an LF50 body composition analyzer after each meal for 28 weeks. Differences between three groups were analyzed using the nonparametric Kruskal-Wallis test. If a statistical difference was detected (*p* < 0.05), post-hoc pairwise group comparisons were performed using Dunn’s test with Bonferroni multiple-testing correction. Differences were considered statistically significantly at *p* < 0.05. *** *p* < 0.001; * *p* < 0.05; ns, not significant.

**Figure 2 foods-09-00688-f002:**
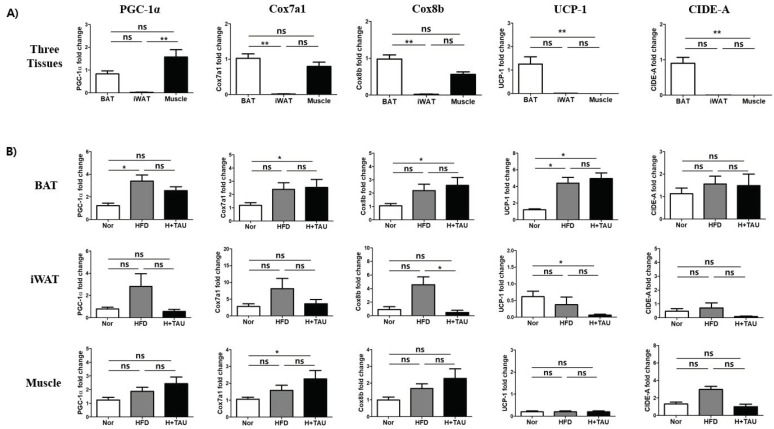
Effects of taurine on transcriptional expression of thermogenesis-related genes in tissues. (**A**) The basal mRNA expression levels of PGC-1α, Cox7a1, Cox8b, UCP-1, and CIDE-A were compared to determine the relative expression of the genes from the three tissues in the normal diet fed (N) group: inguinal white fat tissue (iWAT), interscapular brown fat tissue (BAT), and quadricep muscle tissues. (**B**) The mRNA expression levels of the genes in iWAT, BAT, and muscles of the three groups were compared to determine how HFD and/or taurine supplementation affected the gene expression of each tissue of the three groups; *N* normal diet group (*n* = 5), *HFD* high-fat diet group (*n* = 5), and *H + Tau* HFD + taurine (2% in drinking water) group (*n* = 5). Differences between the three groups were analyzed using the nonparametric Kruskal–Wallis test. If a statistical difference was detected (*p* < 0.05), post hoc pairwise group comparisons were performed using Dunn’s test with Bonferroni multiple-testing correction. The *P* values of the two groups were included to show the expression difference of thermogenesis-related genes. Differences were considered statistically significant at *p* < 0.05. ** *p* < 0.01; * *p* < 0.05; ns not significant. *N* normal diet group, *HFD* high-fat diet group, *H + Tau* HFD + taurine (2% in drinking water) group.

**Figure 3 foods-09-00688-f003:**
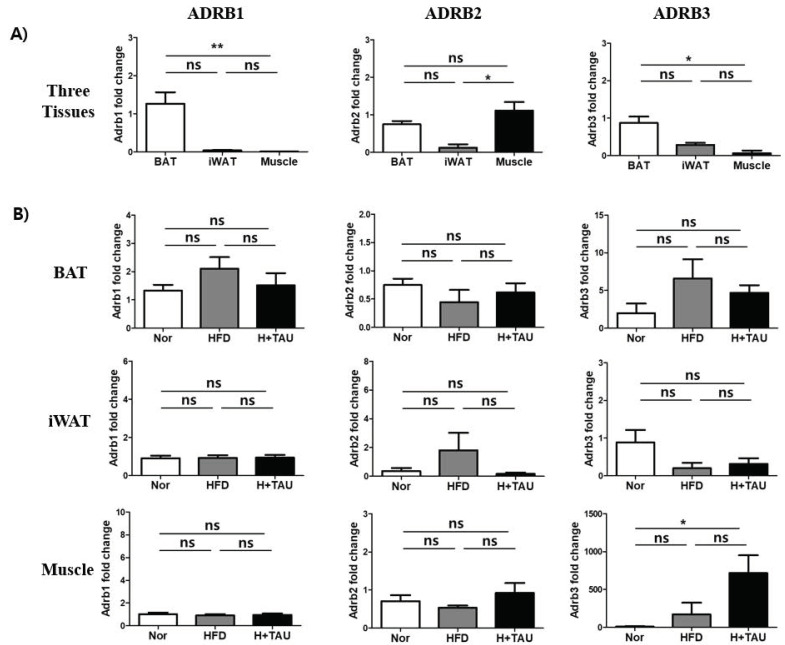
The effects of taurine on the transcriptional expression of β_1, 2, 3_-adrenergic receptors in tissues. (**A**) The relative basal mRNA expression levels of three subtypes of β_1, 2, 3_-adrenergic receptors (*ADRB1*, *ADRB2*, and *ADRB3*) were compared in the three tissues of the normal diet (N) group. (**B**) The mRNA expression levels were compared to determine if HFD and/or taurine supplementation affected gene expression in each tissue of the three groups: *N* normal diet group (*n* = 5), *HFD* high-fat diet group (*n* = 5), and *H + Tau* HFD + taurine (2% in drinking water) group (*n* = 5). Differences between the three groups were analyzed using the nonparametric Kruskal–Wallis test as described in Figure 2. Differences were considered statistically significant at *p* < 0.05. ** *p* < 0.01; * *p* < 0.05; ns not significant.

**Figure 4 foods-09-00688-f004:**
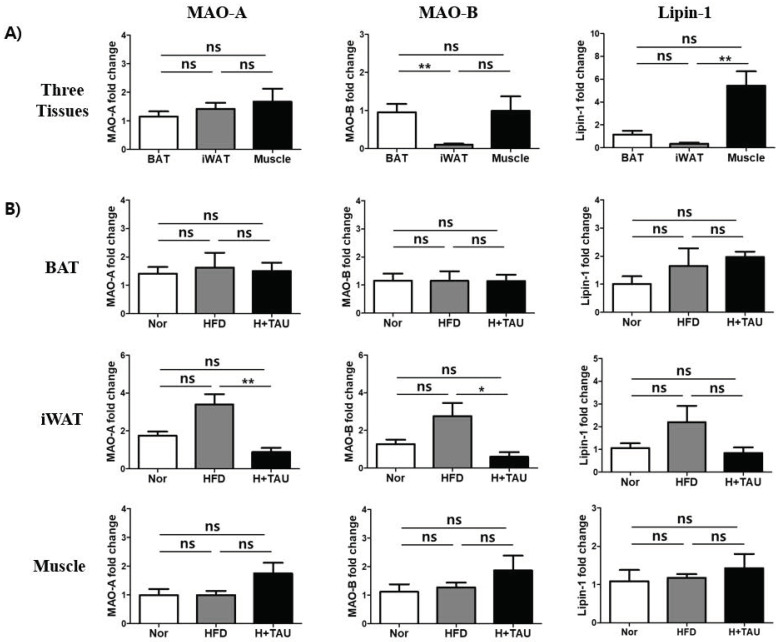
Effects of taurine on the transcriptional expression of monoamine oxidase (MAO) and lipin-1 in tissues. (**A**) The relative basal mRNA expression levels of MAO-A, MAO-B, and lipin-1 were compared in the three tissues of the normal diet (N) group. (**B**) The mRNA expression levels were compared to evaluate the effect of HFD and/or taurine supplementation on each tissue of the three groups: *N* normal diet group (*n* = 5), *HFD* high-fat diet group (*n* = 5), and *H + Tau* HFD + taurine (2% in drinking water) group (*n* = 5). Differences between the three groups were analyzed using the nonparametric Kruskal–Wallis test as described in Figure 2. Differences were considered statistically significant at *p* < 0.05. ** *p* < 0.01; * *p* < 0.05; ns not significant.

**Figure 5 foods-09-00688-f005:**
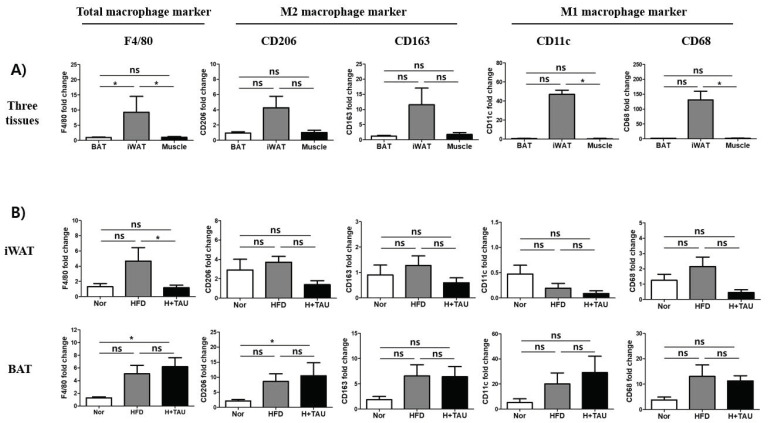
Effect of taurine on the shift from M1 to M2 macrophages in fat tissues. (**A**) The relative basal mRNA expression levels of the macrophage marker genes (F4/80) and M1 (CD11c and CD68) and M2 (CD206 and CD163) macrophage-specific marker genes were investigated to determine how many M1 and M2 macrophages existed in iWAT and BAT of the normal diet (N) group. (**B**) The effect of HFD or taurine supplementation on the mRNA expression levels was investigated to determine if HFD or taurine supplementation affected the infiltration of macrophages into fat tissues and the shift of M1/M2 macrophages in iWAT and BAT: *N* normal diet group (*n* = 5), *HFD* high-fat diet group (*n* = 5), and *H + Tau* HFD + taurine (2% in drinking water) group (*n* = 5). Differences between the three groups were analyzed using the nonparametric Kruskal–Wallis test as described in Figure 2. Differences were considered statistically significant at *p* < 0.05. * *p* < 0.05; ns not significant.

**Table 1 foods-09-00688-t001:** Primer sequences used in the experiment.

PGC-1α	Forward	AGAAGCGGGAGTCTGAAAGG
	Backward	TTCTGTCCGCGTTGTGTCAG
Cox7a1	Forward	CGACAATGACCTCCCAGTACA
	Backward	AGCCCAAGCAGTATAAGCAGTAG
Cox8b	Forward	AAAGCCCATGTCTCTGCCAA
	Backward	TGGAACCATGAAGCCAACGA
UCP-1	Forward	AGTACCCAAGCGTACCAAGC
	Backward	ACCCGAGTCGCAGAAAAGAA
CIDE-A	Forward	AGACCGCCAGGGACTACG
	Backward	GAAACTCGAAAAGGGCGAGC
ADRB1	Forward	ATGGGTGTGTTCACGCTCTG
	Backward	AGAAGACGAAGAGGCGATCC
ADRB2	Forward	AATAGCAACGGCAGAACGGA
	Backward	TCAACGCTAAGGCTAGGCAC
ADRB3	Forward	AAACTGGTTGCGAACTGTGG
	Backward	TAACGCAAAGGGTTGGTGAC
MAO-A	Forward	CGGAAAGCTGAACGACTTGC
	Backward	ACTGCTCCTCACACCAGTTC
MAO-B	Forward	CCCTTGCTGAAGAGTGGGAC
	Backward	TCACAAAGAGCGTGGCAATC
Lipin-1	Forward	ACTGGGAAAGGCCACAATAC
	Backward	GTGCTCTTCATCACTGGAGG
F4/80	Forward	AAGACTGACAACCAGACGGC
	Backward	AAGAGCATCACTGCCTCCAC
CD206	Forward	AGCCTGGAAAGAGCTGTGTG
	Backward	CATCGCTTGCTGAGGGAATG
CD163	Forward	ATGCTTCCATCCAGTGCCTC
	Backward	CTGTCGTCGCTTCAGAGTCC
CD11c	Forward	AGCCTTTCTTCTGCTGTTGG
	Backward	AAATGTGTCGGCTTCTCTGC
CD68	Forward	AAAGGCCGTTACTCTCCTGC
	Backward	ACTCGGGCTCTGATATAGGT

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
