# Peer review of "Taurine Stimulates Thermoregulatory Genes in Brown Fat Tissue and Muscle without an Influence on Inguinal White Fat Tissue in a High-Fat Diet-Induced Obese Mouse Model"

_foods, 2020, doi:10.3390/foods9060688_

Round 1

Reviewer 1 Report

I read your manuscript with a great interest. However several points need to be improved :

1) in the intoduction section, browing, adipose tissue has to be  develop with adequate refernces. The genes you analyzed later also have to be presented, as their roles in browning, thermogenesis ... explained with appropriate recent references

In the materila and methods section. we need to understand :

1) why were the experiments performed only on males ?

2) What is the composition of HFD, specially its percentages in saturated fatty acids, unsaturated ones and w6/w3 ratio. We need the same informations the for chow diet.

3) taurine is 2% in drinking water. But how did you make sure that all the animals drank the seame amount ? Were they house in individual cages? What amount of taurine per gram of weight body did they receive?

In the results section :

Fig 2 A : in the figure itself, add "normal diet" under "three tissues". Same for 3A

Fig2 and 3 : display fold changes and not “mRNA levels”.

Remove last 2 lines of page 9 “in contrast, taurine supplementation slightly decreased …) as this is not statistically significant. Or rewrite sentence as it is significant for Cox8b and UCP1

Page 11 : provide reference demonstrating that β1,2,3 adrenergic receptors promotes beiging.

Page 13, last 2 lines : there are not enough of results to claim that taurine acts through the mechanisms you describe. It is only an indication.

Same thing page 15 : only F4/80 variations in iWAT are significant. It could also be the result of a lower expression of this gene on the same number of macrophages

The discussion section needs to be greatly improved, that will be possible as far as th introduction will also have been.

At this stage, your title is over-interpretation of your results. Taurin reduces the deleterious effects of HFD. No more. You can also not say that it is safe as you did not provide any infromation about the animals'behavior, or any toxicological results.

Author Response

I read your manuscript with a great interest. However several points need to be improved :

1) in the introduction section, browning, adipose tissue has to be develop with adequate references. The genes you analyzed later also have to be presented, as their roles in browning, thermogenesis ... explained with appropriate recent references

Response: The authors thank the reviewer for their suggestions. We have now added appropriate reference regarding BAT and brite included as reference #26. Various genes and their roles in browning, thermogenesis have been explained in the Discussion section of the revised manuscript.

In the materials and methods section. we need to understand :

1) why were the experiments performed only on males ?

Response: The rationale for choosing males for conducting the experiments in this study has been based on previously published papers where males have been used for these kind of experiments. Additionally, there is also the issue of the female estrus cycle that possibly could interfere with the data. The authors thank the reviewer for pointing this out.

2) What is the composition of HFD, specially its percentages in saturated fatty acids, unsaturated ones and w6/w3 ratio. We need the same information the for chow diet.

Response: The HFD, Research Diets D12451 diet (45 kcal % fat), was purchased from Nara Biotech (Seoul, Korea). Unfortunately, there is no information about the ratio of W6/W3. The authors would like to apologize in this regard. It is known that w6 induces more severe obesity than w3 does. We thank the reviewer for this relevant question.

3) taurine is 2% in drinking water. But how did you make sure that all the animals drank the same amount ? Were they house in individual cages? What amount of taurine per gram of weight body did they receive?

Response: The authors have performed similar experiments previously, wherein the mentioned parameters were checked for 48 hrs. in metabolic cages. The reviewer might kindly check the reference: Anti-obesity effect of taurine through inhibition of adipogenesis in white fat tissue but not in brown fat tissue in a high-fat diet-induced obese mouse model. Amino Acids 2019, 51, (2), 245-254.

In the results section :

Fig 2 A : in the figure itself, add "normal diet" under "three tissues". Same for 3A

Response: The authors have described this in the figure legends due to the lack of space in fitting a large label in the figure itself. The authors apologize for not including it in the figure.

Fig2 and 3 : display fold changes and not “mRNA levels”.

Response: The authors have made necessary corrections in the revised manuscript as per the reviewer’s recommendations.

Remove last 2 lines of page 9 “in contrast, taurine supplementation slightly decreased …) as this is not statistically significant. Or rewrite sentence as it is significant for Cox8b and UCP1.

Response: As per the reviewer’s suggestions the specified lines have been removed in the revised manuscript.

Page 11 : provide reference demonstrating that β1,2,3 adrenergic receptors promotes beiging.

Response: Appropriate references have now been included in the revised manuscript.

Page 13, last 2 lines : there are not enough of results to claim that taurine acts through the mechanisms you describe. It is only an indication.

Response: In order to manuscript read for clearly, the authors have changed the sentence to the following, “These results indirectly suggest that taurine supplementation partly contributes to anti-obesity effect via the inhibition of lipid synthesis in iWAT but not in BAT or muscle”. Thank you for your critical comments.

Same thing page 15 : only F4/80 variations in iWAT are significant. It could also be the result of a lower expression of this gene on the same number of macrophages

Response: F4/80 is one of macrophage’s markers. The expression may be altered but in the normal tissue the expression level usually indicates the number of macrophages. Thank you.

The discussion section needs to be greatly improved, that will be possible as far as the introduction will also have been.

Response: The authors thank the reviewer for their critical comments. The discussion section has been slightly change in the revise manuscript.

At this stage, your title is over-interpretation of your results. Taurine reduces the deleterious effects of HFD. No more. You can also not say that it is safe as you did not provide any information about the animals'behavior, or any toxicological results.

Response: The authors greatly appreciate the reviewer’s concerns regarding the safety of taurine administration. However, Taurine has been used for a long time without any reported toxic effects. It is a very safe component. Studies have only reported taurine toxicity only if it is used at high dose. Thank you.

Reviewer 2 Report

This manuscript by Kim and colleagues aims to evalute expression profile of thermogenic genes in muscles and adipose tissue using an animal model of obesity. Specifically, they investigated the effect of 28-days supplementation with 2% taurine. In my opinion, this manuscript and its potential perspectives are interesting. However, I have the following comments along with a suggestion to revise the text for typos and confusing paragraph.

In the abstract, I would suggest to avoid general sentences. I prefer to read what are the main comparisons and their main findings.

Introduction and Methods are appropriate and clear….

In the results section, and especially in Figure legends, I noted discrepancy in the Number of animals analyzed. The Authors stated that each group consisted in ten animals. However, in Fig 1, 2,3, 4 I found different numbers.

In the discussion section, I suggest to remove the sentence "This suggests that long term taurine intake at optimal doses in the day to day life may help
reduce body weight in human with mild obesity."

Author Response

This manuscript by Kim and colleagues aims to evaluate expression profile of thermogenic genes in muscles and adipose tissue using an animal model of obesity. Specifically, they investigated the effect of 28-days supplementation with 2% taurine. In my opinion, this manuscript and its potential perspectives are interesting. However, I have the following comments along with a suggestion to revise the text for typos and confusing paragraph.

In the abstract, I would suggest to avoid general sentences. I prefer to read what are the main comparisons and their main findings.

Response: The authors agree with the reviewer’s comments. However, in this study, the effect of taurine on various gene expression was studied. Therefore, the abstract may seem to be described broadly. The authors hope the reviewer will understand our perspective. We thank you for your critical comment.

Introduction and Methods are appropriate and clear….

In the results section, and especially in Figure legends, I noted discrepancy in the Number of animals analyzed. The Authors stated that each group consisted in ten animals. However, in Fig 1, 2,3, 4 I found different numbers.

Response: At the onset of our experimental study, each group consisted of 10 mice. There were shortages in the availability of metabolic cages. Gene analysis was also not available for all the samples. We hope the reviewer can understand the situation.

In the discussion section, I suggest to remove the sentence "This suggests that long term taurine intake at optimal doses in the day to day life may help reduce body weight in human with mild obesity."

Response: The authors have removed the sentence in the revised manuscript as per the reviewer’s suggestion.

Reviewer 3 Report

General Comments:

In my opinion, it’s a valuable topic; the manuscript is well written.

The figures are well built, clean. Describing footnote information is appropriate.

The following are my comments to the authors.

Introduction:

  • Second paragraph: please provide citation regarding taurine effects/ deficiency associated with diabetes.

  • Second paragraph: I would like to read more recent citations, especially from the last 5 years.

Please review: The effects of taurine supplementation on oxidative stress indices and inflammation biomarkers in patients with type 2 diabetes: a randomized, double-blind, placebo-controlled trial. Diabetol Metab Syndr. 2020 Jan 29;12:9. doi: 10.1186/s13098-020-0518-7. eCollection 2020.

It may be useful to include.

  • Please, check: “One of the molecular mechanisms by which taurine ameliorates mild obesity, in our study, we showed that taurine supplementation downregulated the expression of adipogenesis-related genes such as PPAR-α, PPAR-γ, C/EBP-α, C/EBP-β, and AP2 in white adipose tissue (WAT) but not in brown adipose tissue (BAT)”

Maybe rephrase the paragraph? In our study, we showed one of the molecular …

Materials and Methods:

I do not have open access to the previous published study. Please address details as follow:

  • Were rats kept individually?
  • How was every-day water consumption assessed?
  • Was the unconsumed water remaining in the feeder take into account?
  • If the allowed extension for the manuscript has not been reach, please provide very brief details about the chow diet.

Results:

  • It would be suitable to include p-values (p< xx) in this section, not only in figures; this inclusion facilitates to distinguish significant results; however, I’ll leave it to author’s preferences.
  • Well written, great structure and easy to follow.

Discussion:

  • Second paragraph: please correct citation style of Bonet et al. 2017. It should be consistent with all others.
  • Well written, well structured.

Any limitations?

Author Response

General Comments:

In my opinion, it’s a valuable topic; the manuscript is well written.

The figures are well built, clean. Describing footnote information is appropriate.

The following are my comments to the authors.

Introduction:

Second paragraph: please provide citation regarding taurine effects/ deficiency associated with diabetes.

Response: Citations have been provided in the revised manuscript as recommended by the reviewer.

Second paragraph: I would like to read more recent citations, especially from the last 5 years.

Response: Unfortunately, not many studies have been conducted in very recent times. Most of the studies were done before 5 years. However, some references have been added in the revised manuscript. Thank you.

Please review: The effects of taurine supplementation on oxidative stress indices and inflammation biomarkers in patients with type 2 diabetes: a randomized, double-blind, placebo-controlled trial. Diabetol Metab Syndr. 2020 Jan 29;12:9. doi: 10.1186/s13098-020-0518-7. eCollection 2020.

It may be useful to include.

Response: The authors would like to thank the reviewer in helping improvise our manuscript. The above reference has been included in the revised manuscript.

Please, check: “One of the molecular mechanisms by which taurine ameliorates mild obesity, in our study, we showed that taurine supplementation downregulated the expression of adipogenesis-related genes such as PPAR-α, PPAR-γ, C/EBP-α, C/EBP-β, and AP2 in white adipose tissue (WAT) but not in brown adipose tissue (BAT)”

Maybe rephrase the paragraph? In our study, we showed one of the molecular …

Response: The authors thank the reviewer for their insightful comments. The authors have incorporated the corrected the sentence.

Materials and Methods:

I do not have open access to the previous published study. Please address details as follow:

Were rats kept individually?

Response: Yes.

How was every-day water consumption assessed?

Response: No

Was the unconsumed water remaining in the feeder take into account?

Response: No

Response: Metabolic monitoring was assessed in a resting state using the PhenoMaster System (TSE systems GmbH, Bad Homburg, Germany). Energy expenditures including CO2 production (VCO2) and O2consumption (VO2) were monitored for 48 h. The mice were free to consume food and water. The respiratory exchange ratio (RER) was defined as the ratio of carbon dioxide volume versus oxygen volume (VCO2/VO2). Food uptake and locomotor activity were also measured.

If the allowed extension for the manuscript has not been reach, please provide very brief details about the chow diet.

Response: The normal diet was purchased from Orient Bio (Korea). The HFD, Research Diets D12451 diet (45 kcal % fat), was purchased from Nara Biotech (Seoul, Korea).

Results:

It would be suitable to include p-values (p< xx) in this section, not only in figures; this inclusion facilitates to distinguish significant results; however, I’ll leave it to author’s preferences.

Well written, great structure and easy to follow.

Response: The authors would like to thank the reviewer for their suggestions. Thank you.

Discussion:

Second paragraph: please correct citation style of Bonet et al. 2017. It should be consistent with all others.

Response: The authors thank the reviewer for pointing out the error. Necessary correction has been made in the revised manuscript.

Well written, well structured.

Round 2

Reviewer 1 Report

The modifications I asked were not performed.

There are only slight modifications but no in deep ones. Over-interpretations remains.

For instance in the introduction I asked more inforamtions about the genes that are further studied. The answer is a single sentence and no desciptpion of the roles that are claimed later on to regulate browning.

The readers shall not have to find the previous publication and read the material and methods section to find informations about the amount of taurine the animals effectively drank.

Same remark for the results : it is not because in normal tissue F4/80 is one of macrophage’s markers that its  expression level may not be altered  under pathological conditions.

Author Response

The modifications I asked were not performed.

There are only slight modifications but no in deep ones. Over-interpretations remains.

For instance in the introduction I asked more inforamtions about the genes that are further studied. The answer is a single sentence and no desciptpion of the roles that are claimed later on to regulate browning.

Response: The authors have now modified the information about thermoregulatory genes in the introduction section of the revised manuscript. A more detailed description of the genes has been relocated from Results and Discussion section to the Introduction.

The readers shall not have to find the previous publication and read the material and methods section to find informations about the amount of taurine the animals effectively drank.

Response: The authors understand the reviewers concern to include specific information regarding the amount of taurine intake. However, the exact data and information for that is not available at the moment. As described in Methods, all the metabolic data were obtained from metabolic cages for only 48 hrs.

The authors initially designed the study based on previously published research that have implemented similar methodology. Recent publications following a similar methodology for taurine supplementation have also not mentioned or defined the exact amount of taurine intake.

Some of the papers are as follows:

Freitas, Israelle Netto, Thiago Reis Araujo, Jean Franciesco Vettorazzi, Emily Amorim Magalhães, Everardo Magalhães Carneiro, Maria Lúcia Bonfleur and Rosane Aparecida Ribeiro. “Taurine supplementation in high-fat diet fed male mice attenuates endocrine pancreatic dysfunction in their male offspring.” Amino Acids 51 (2019): 727-738.Guo, Ying-Ying, Bai-Yu Li, Wan-Qiu Peng, Liang Guo, and Qi-Qun Tang. "Taurine-mediated browning of white adipose tissue is involved in its anti-obesity effect in mice." Journal of Biological Chemistry 294, no. 41 (2019): 15014-15024.

The authors have however added more information about metabolic cages that animals’ behavior, food uptake and drinking in the Materials and Methods section of the revised manuscript.

Same remark for the results : it is not because in normal tissue F4/80 is one of macrophage’s markers that its expression level may not be altered under pathological conditions.

Response: The authors apologise for misunderstanding the question earlier. The expression is not changed but more macrophages are infiltrated into iWAT. The authors have now corrected the sentence in the revised manuscript. Thank you.

Reviewer 2 Report

None

Author Response

The authors edited and checked the manuscript again. Thank you.